



# Response of ice sheets, sea-ice and sea level in climate stabilisation and reversibility simulations using a state-of-the-art Earth System Model

Robin S. Smith[1], Tarkan A. Bilge[2], Thomas J. Bracegirdle[2], Paul R. Holland[2], Till Kuhlbrodt[1], Charlotte Lang[1], Spencer Liddicoat[3], Tom Mitcham[4], Jane Mulcahy[3], Kaitlin A. Naughten[2], Andrew Orr[2], Julien Palmieri[5], Antony J. Payne[6], Steve Rumbold[1], Marc Stringer[1], Ranjini Swaminathan[7], Sarah Taylor[2], Jeremy Walton[3], and Colin Jones[8]

[1]National Centre for Atmospheric Science, University of Reading, Reading, UK
[2]British Antarctic Survey, Cambridge, UK
[3]Met Office Hadley Centre for Climate Science and Services, Exeter, UK
[4]Centre for Polar Observation and Modelling, University of Bristol, Bristol, UK
[5]National Oceanography Centre, Southampton, UK
[6]Department of Earth, Ocean and Ecological Sciences, University of Liverpool, Liverpool, UK
[7]National Centre for Earth Observation, University of Reading, Reading, UK
[8]National Centre for Atmospheric Science, University of Leeds, Leeds, UK

**Correspondence:** Robin S. Smith (robin.smith@ncas.ac.uk)

**Abstract.** We have conducted an ensemble of idealised climate overshoot simulations in a state-of-the-art Earth system model in which global mean temperature is increased at a constant rate to various global warming levels (GWLs) by prescribing constant $CO_2$ emissions, followed by a period of zero $CO_2$ emissions started at different GWLs, and then a period of cooling in which $CO_2$ is removed from the atmosphere. We give an overview of the ice sheet, sea-ice and sea level responses in these simulations highlighting long term responses at different GWLs and discuss the degree to which those responses can be
simply reversed as global surface temperature cools. We show a broad divide between the two hemispheres, in which northern hemisphere polar processes have larger direct responses to warming which are more simply reversible than those in the southern hemisphere.

Cessation of $CO_2$ emissions at most GWLs stabilises surface temperatures at high northern latitudes, although a slow warm-
ing trend continues at high southern latitudes. Northern hemisphere sea-ice extent and Greenland surface mass balance both stabilise under zero $CO_2$ emissions and appear to return toward preindustrial levels along with surface temperatures when $CO_2$ is sequestered, but southern hemisphere sea-ice continues to decline under zero emissions and does not simply increase again as global temperatures cool. Likewise, the Antarctic circumpolar westerlies exhibit strengthening and shift poleward under global warming, but do not simply return to their preindustrial state when the climate is cooled, with implications for ocean
circulation and Antarctic surface mass balance. The thermosteric contributions to global mean sea level from ocean heat uptake and the Greenland ice sheet continue when $CO_2$ emissions are ceased or reversed, at rates which slowly decline on centennial timescales. The net mass balance of Antarctica and its contribution to sea level do not simply scale with global temperature;



they result from a complex interaction between the basal and surface mass balances, the ice-dynamic response to those forcings and significant trends inherent to the initialisation of our ice sheet model state.

## 1 Introduction

Changes in the Earth system at high latitudes are some of the largest and most impactful features of anthropogenically forced climate change. Polar amplification of surface temperature change means the Arctic has warmed nearly four times faster than the global average in recent decades (Rantanen et al., 2022), with consequent impacts on physical (Walsh et al., 2020) and biological (Myers-Smith et al., 2020) systems. Arctic sea-ice has rapidly declined over recent decades (Stroeve and Notz, 2018), with profound impacts on ecology (Post et al., 2013), human activity (Meier et al., 2014), and Eurasian weather (Vihma, 2014). Antarctic sea-ice has been declining since 2016, resulting in recent record-low sea-ice cover (Purich and Doddridge, 2023). The Greenland and Antarctic ice sheets and shelves are losing mass due to warming in the atmosphere and ocean (Mouginot et al., 2019; Naughten et al., 2022), contributing significant amounts of fresh water to the oceans impacting regional oceanography and global sea levels.

There is significant uncertainty in how these features of the Earth system will evolve as the climate changes further. Different methods of projecting the future evolution of climate change disagree about the timing of sea-ice-free conditions in the Arctic, how significantly the Atlantic Meridional Overturning Circulation (AMOC) will decline (or even collapse), the importance of carbon cycle feedbacks associated with permafrost thaw and Arctic greening and how rapidly sea levels will rise (IPCC, 2022). Some of these systems include feedbacks that might lead to non-linear abrupt shifts in their behaviour rather than gradual changes, or to changes that would not be reversed if surface temperatures were allowed to warm beyond certain thresholds before cooling again (Armstrong McKay et al., 2022). Given the potential for very significant impacts from high latitude Earth system change, it is crucial we improve our understanding of how these systems will respond to different levels and rates of greenhouse gas forcing.

To investigate knowledge gaps related to the transient behaviour of different Earth system components in overshoot scenarios, we have conducted an ensemble of simulations with a state-of-the-art Earth System Model (ESM) capable of representing many of the key interactions between the physical climate system, the global carbon cycle, vegetation and the Earth's major ice sheets. There is some literature on the behaviour of individual aspects of the Earth system as atmospheric $CO_2$ concentrations ramp up and back down (e.g. Boucher et al., 2012; Jackson et al., 2013; Hwang et al., 2024), but these simulations have not been performed in a systematic way, nor with such a sophisticated ESM that includes a closed land and ocean carbon cycle driven by anthropogenic greenhouse gas emissions rather than specified atmospheric $CO_2$ concentrations. Many of the closest potential Earth system tipping points highlighted by Armstrong McKay et al. (2022) and others involve the ice sheets of Greenland and Antarctica, so it is particularly important that our model also includes sophisticated, two-way coupled representations of these ice sheets and their interactions with the atmosphere and ocean (Smith et al., 2021; Siahaan et al., 2022).

This paper describes the response of the polar climate, ice sheets and sea-ice components in our ensemble of simulations. The results of this large modelling exercise are extremely rich in detail, so in this initial publication we choose to give a



broad, top-level account of their evolution; more process-level detail of the key phenomena, including analysis of non-polar components, will follow in subsequent publications. The paper is structured as follows: in Methods (section 2) we briefly describe the ESM being used and how the overshoot ensemble is designed. In Results (section 3) we first give an overview of the response of global mean surface temperature, atmospheric $CO_2$ concentrations and ocean heat content changes before looking at the behaviour of northern hemisphere sea-ice, the Greenland ice sheet, southern hemisphere sea-ice and westerly winds, and the Antarctic ice sheet. We conclude with a discussion (section 4) of the implications for reversibility of cryosphere components of the Earth system and sea level rise and discuss further analysis which will come based on this ensemble of simulations.

## 2 Methods

### 2.1 Experiment Protocol

We use an experiment protocol in which global mean surface air temperatures (GSAT) are increased at a fixed rate of 2°C/century through a specified emission of $CO_2$ into the model atmosphere. At specified GWLs, $CO_2$ emissions are set to zero, with a view to stabilising global warming. Fifty years into each zero-emission run (at different GWLs) $CO_2$ emissions become negative and the climate is cooled back to preindustrial temperatures. This idealised framework allows Earth system changes to be expressed as a response to a simple global forcing. The protocol can be readily used across different ESMs, whereby different $CO_2$ emission rates (based on each model's TCRE (transient climate response to cumulative emissions of $CO_2$) value) deliver the same global warming rate, with zero and negative emission runs spawned at the same GWLs across models. This experiment protocol forms the TIPMIP ESM Tier 1 experiment protocol (Winkelmann et al., 2025; Jones and al., 2025). It is important to note that changes in GSAT are realised via idealised $CO_2$ emissions and sequestration alone. Neither aerosol forcing, trace gas emissions, or any other greenhouse gas or anthropogenic boundary condition change form part of this idealised framework. In this paper we deviate from the formal TIPMIP naming convention when referring to our experiments on grounds of readability; a mapping between the names used here and the formal TIPMIP ESM designations can be found in Appendix A.

The control simulation represents an indefinite preindustrial climate, with no anthropogenic $CO_2$ emission (experiment "PI"). Specifications for boundary conditions come from the CMIP6 piControl experiment (Eyring et al., 2016). The initial conditions for this simulation are derived from the UKESM1.1 concentration-driven preindustrial simulation (Mulcahy et al., 2023) which has spun up long enough for global mean surface air temperatures to be stable, and for fluxes into and out of the model carbon sources and sinks to be in balance, so global mean atmospheric $CO_2$ does not drift when allowed to evolve prognostically (Fig. 1b,c).

GSAT increases from the PI baseline under a constant $CO_2$ emission of 8GtC/yr (experiment "Up8"). This emission rate was chosen to produce a warming rate of ~2°C/century in UKESM, roughly equivalent to the rate of warming seen in both the CMIP6 *1pctCO2* experiment and observations of the global warming rate over the past half-century (Lenssen et al., 2024).





The emissions have the same spatial pattern as used in the final year of the CMIP6 $CO_2$ Historical experiment (Hoesly et al., 2018), scaled to give a constant emission rate of 8GtC/yr.

As the Up8 simulation proceeds, when GSAT anomalies with respect to PI reach certain thresholds, our simulations attempt to stabilise at that global warming level (GWL) by cutting the $CO_2$ emission rate to zero again (experiments "ZE-$x$", for x=[1.5,2,3,4,5] °C). Net zero $CO_2$ emissions are not guaranteed to produce zero temperature trends in an Earth system simulation of this sort, as ongoing removal of excess $CO_2$ from the atmosphere by the land and ocean carbon sinks provides a cooling tendency whilst opposing factors such as the adjustment of the ocean heat budget tend to lead to surface air warming

(MacDougall et al., 2020). Initial testing with UKESM suggested that in this idealised framework these two tendencies largely cancel out, such that zero $CO_2$ emission produces a reasonably stable GSAT (at least for GWLs at or below 2°C, Fig 1c).

     To represent possible overshoot pathways of GSAT in our protocol and investigate the reversibility of simulated changes, other simulations allow the climate to stabilise at a GWL for some time before reducing GSAT again. This cooling is achieved by removing $CO_2$ from the atmosphere, reversing the sign of $CO_2$ emissions to simulate net sequestration of $CO_2$ from the

lower atmosphere. The simulations in this paper stabilise (i.e. run with zero emissions of $CO_2$) for 500 years at each GWL. Fifty years into each zero-emission run a negative emission run is branched off, with emissions set to -4GtC/yr, half of the rate used during the ramp-up (experiments "Dn4-$x$", for each GWL $x$). This $CO_2$ sink has the same spatial pattern as the $CO_2$ source used in the ramp-up.

     The TIPMIP ESM protocol (Jones and al., 2025) requests zero-emission runs at GWL = 2°C and 4°C, with negative emission

runs back to PI conditions branched from these after 50 years. We extend the TIPMIP ESM protocol, sampling zero-emission runs at additional GWLs, different rates of negative $CO_2$ emissions started 50 and 200 years into each zero-emission run, and multiple ensemble members for each experiment. This allows a more complete analysis of the warming-stabilisation-cooling (overshoot) space. In the interests of brevity, in this overview paper our results are drawn from one member of each warming, stabilisation and cooling phase for each GWL, using the stabilisation period and sequestration rate noted above. The evolution

of the simulations shown can be taken as generally representative of the major features of all related variants at each GWL in the full UKESM ensemble. The full ensemble behaviour will be extensively analysed in subsequent papers.

## 2.2   Model

We use UKESM1.2, a version formed by combining UKESM1.1 as described in Mulcahy et al. (2023) and Sellar et al. (2019), with the two-way coupled Greenland and Antarctic ice sheet components described for variants of UKESM1 in Smith et al.

(2021) and Siahaan et al. (2022). UKESM1.2 uses the N96 resolution (1.875° longitude 1.25° latitude) of the Unified Model atmosphere component version 3.1 (Kuhlbrodt et al., 2018; Williams et al., 2018) with 85 vertical levels extending to the mesosphere. For the ocean, we use the G07 version of NEMO (Storkey et al., 2018; Madec et al., 2017) on an ORCA1 (equivalent 1° resolution) ocean grid with 75 vertical levels, including explicit modelling of the ocean and melt under floating Antarctic ice shelves. The land surface and terrestrial carbon cycle is modelled by JULES (Best et al., 2011; Clark et al., 2011)

using 13 plant functional types. Our configuration of JULES includes an online downscaling of ice sheet surface exchange based on sub-gridscale elevation classes and a multilayer snow model that allows for percolation and refreezing of liquid water





within the snowpack. The distribution of vegetation is interactively calculated by the TRIFFID model (Cox, 2001). We use a complex, interactive atmospheric chemistry and aerosol scheme provided by UKCA (Mulcahy et al., 2018; Archibald et al., 2020) and ocean biogeochemistry from MEDUSA (Yool et al., 2013) which uses five marine functional types. The Greenland

and Antarctic ice sheets are simulated by BISICLES (Cornford et al., 2015), a vertically integrated model of ice-sheet dynamics which uses an adaptive mesh with a horizontal resolution down to ∼1km to accurately model processes such as grounding line movement. We refer readers to the listed references for a complete description of each of these components and how they are coupled together.

## 2.3 Baseline simulation and ice sheet initialisation

This ensemble uses a baseline simulation with constant preindustrial forcing of indefinite duration, similar to that defined by the CMIP6 protocol, from which the Up8 simulation branches. For an ESM with interactive ice sheets, this concept raises two issues. Firstly, there is insufficient evidence of what the preindustrial states of the Greenland and Antarctic ice sheets (GrIS, AIS) actually were, importantly including a lack of information on the position of the grounding lines of key outlet glaciers. The sensitivity of the ice sheet response to changes in boundary conditions is known to depend on the details of the initial

state (Goelzer et al., 2018; Seroussi et al., 2019). Moreover, the concept of the CMIP6 preindustrial control simulation assumes that the key metrics of the Earth system in which we are interested are in some quasi-stable equilibrium with the preindustrial forcing. In reality, the GrIS and AIS in the preindustrial would have been transiently evolving away from their Last Glacial Maximum states at some rate. Within our imperfect modelling systems, it is impossible to realise a preindustrial simulation with indefinitely stable ice sheet states that are simultaneously realistic and in balance with the biases inherent to a given ESM.

Developing best-practice approaches to the problem of ice sheet initialisation in global ESM simulations is an area of active research that we do not attempt to address here. In the simulations described in this paper, as in previous UKESM simulations with interactive ice sheets, we use the initialisations of the GrIS and AIS developed to represent the states of the ice sheets at the beginning of the $21^{st}$ century (Lee et al., 2015; Cornford et al., 2015). This was felt to be the simplest and cleanest approach at the time of initialising our ensemble. Preliminary attempts to grow our simulated ice sheets from their modern states to

expanded states that might be more appropriate for the preindustrial and use these to initialise $20^{th}$ century simulations have so far resulted in biases in ice thickness and velocity that are worse than those that arise from simply starting from the modern states. During the baseline PI simulation we do not allow the ice sheet geometry to evolve away from those modern states, although we do interactively compute the surface mass balance (SMB) and basal mass balance (BMB) at the ice sheet surfaces that result from their presence in the preindustrial climate and pass the resultant freshwater fluxes to the ocean. When the Up8

simulation is branched from PI the ice sheet geometries are then allowed to start evolving, and this evolution then continues through all subsequent ZE-*x* and Dn4-*x* branches.

One of the key issues with using a modern initialisation of the ice sheets when starting our simulations from preindustrial conditions is that the modern BISICLES AIS initialisation includes grounding lines for key ice streams in the West AIS that are already retreating (e.g. Pine Island Glacier and Thwaites Glacier). This retreat is not halted by applying UKESM preindustrial

boundary conditions to the ice sheet, so that our realisation of this sector of the ice sheet is effectively already past a tipping





point (Bett et al., 2024) with respect to the UKESM climatic forcing. As a result, every AIS simulation in our ensemble has a historically recent rate of ice loss built into it from the beginning, a factor that we take into consideration in our analysis. To enable this analysis, we conduct an 'ice control' simulation that allows the ice sheet geometry to evolve under continued preindustrial boundary conditions to isolate the unforced behaviour of our AIS component (experiment "ZE-0").

## 3 Results

### 3.1 Global Surface Temperature

Although the focus of this paper is on polar climate and ice sheets, for context we first present the basic global surface temperature responses of UKESM under the TIPMIP ESM protocol.

The accumulating burden of emitted $CO_2$ in UKESM illustrates the positive emission (warming), zero-emission (stabilisa-
tion) and negative emission (cooling) phases of the forcing protocol we are using (Fig. 1a). The concentration of $CO_2$ in the atmosphere increases in Up8 and then slowly drops in the ZE-$x$ simulations when emissions cease as the excess $CO_2$ is taken up by the land and ocean, with the rate of uptake being slightly higher for experiments that have stabilised at higher GWLs (i.e. with greater emitted $CO_2$, Fig 1b). Sequestration of $CO_2$ adds strongly to the effect of the natural carbon sinks and $CO_2$ concentrations drop rapidly in the Dn4-$x$ simulations.

Global mean air surface temperature broadly follows the atmospheric $CO_2$ burden (Fig. 1c). Despite the slow fall in atmospheric $CO_2$ in the ZE-$x$ simulations, surface temperature trends are near zero for GWLs at or below 2°C, and slightly positive for higher GWLs. This comes about from a balance between the surface warming trend associated with heat sequestered into the deep ocean reemerging at the surface as the climate comes into equilibrium, and cooling from the slow decrease in atmospheric $CO_2$ in ZE-$x$ (Gibbs and al., 2025). Surface temperatures generally cool in line with the rate of $CO_2$ removal in the
Dn4-$x$ simulations as radiative forcing from excess $CO_2$ in the atmosphere is rapidly reduced. At GWLs less than 4°C there is a consistent relationship between GSAT and the accumulated amount of $CO_2$ in the system, regardless of whether $CO_2$ is being emitted or sequestered (Fig. 1d). In the cooling Dn4-4 and Dn4-5 GSAT is slightly higher for any given amount of accumulated $CO_2$ than it was for that burden as Up8 warmed, reflecting warming in the ocean that cannot be as rapidly lost when atmospheric $CO_2$ is reduced.

Surface air temperature (SAT) change at high latitudes is usually greater than the global mean due to several polar amplification mechanisms, especially in the northern hemisphere (Goosse et al., 2018). In our simulations Arctic SAT is a factor of two to three times higher than the global average (Fig. 2a,c). The amplification factor is highest for zero-emission runs at lower GWL simulations, likely because at higher GWLs there is less Arctic sea-ice and high latitude snow available to drive a positive ice albedo feedback and insulate the atmosphere from the warmer ocean. For the highest GWLs the amplification
factor declines with time because GSAT slowly increases in these runs whilst Arctic SAT itself remains constant. The warming trend seen in GSAT in ZE-$x$ simulations at higher GWLs is largely driven by warming at high southern latitudes (Fig. 2b,d), which have a lower amplification factor than the northern hemisphere SAT. This warming, which is especially marked at higher GWLs, is due to the reemergence in the Southern Ocean upwelling zones of heat that has been taken up by the ocean during the




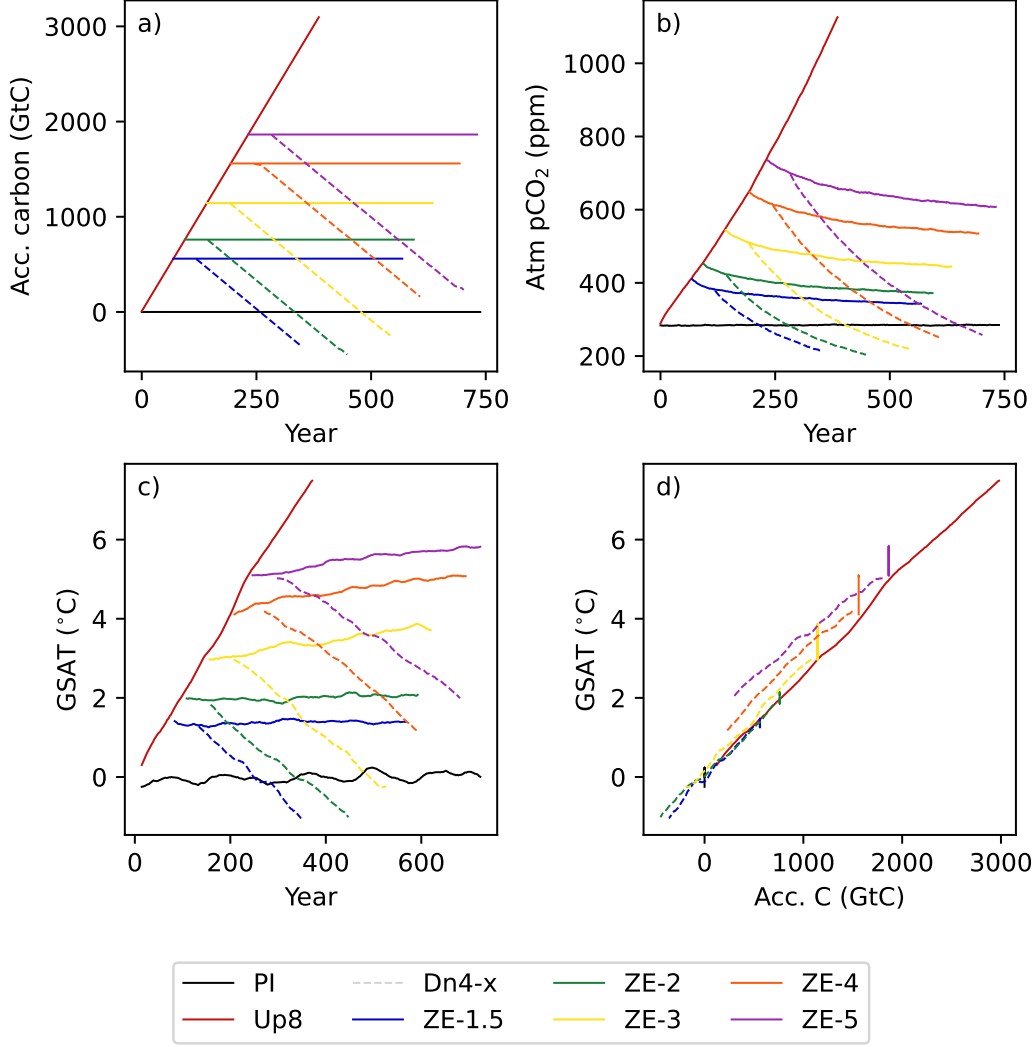

**Figure 1.** Time evolution of a) accumulated inventory of $CO_2$ emitted into the Earth system, b) concentration of $CO_2$ in the atmosphere, c) anomaly in global mean surface air temperature (GSAT) with respect to the time mean of PI, d) variation of GSAT with inventory of emitted $CO_2$. All quantities are smoothed with a 30-year running mean. Solid lines show PI, Up8 and ZE-$x$ simulations, dashed lines of the same colour show Dn4-$x$ cooling simulations from each GWL.

warming phase of the protocol and sequestered below the thermocline. The reemergence of this heat at the surface additionally
triggers positive feedbacks in Antarctic sea-ice melt and reduces local cloud fraction which both act to amplify the surface heating.

There is a large multi-decadal warming in southern hemisphere SAT of around 2°C in the Dn4-3 simulation around when temperatures have returned to PI levels (Fig. 2b). This appears to be due to an episode of rapid sea-ice loss and surface warming



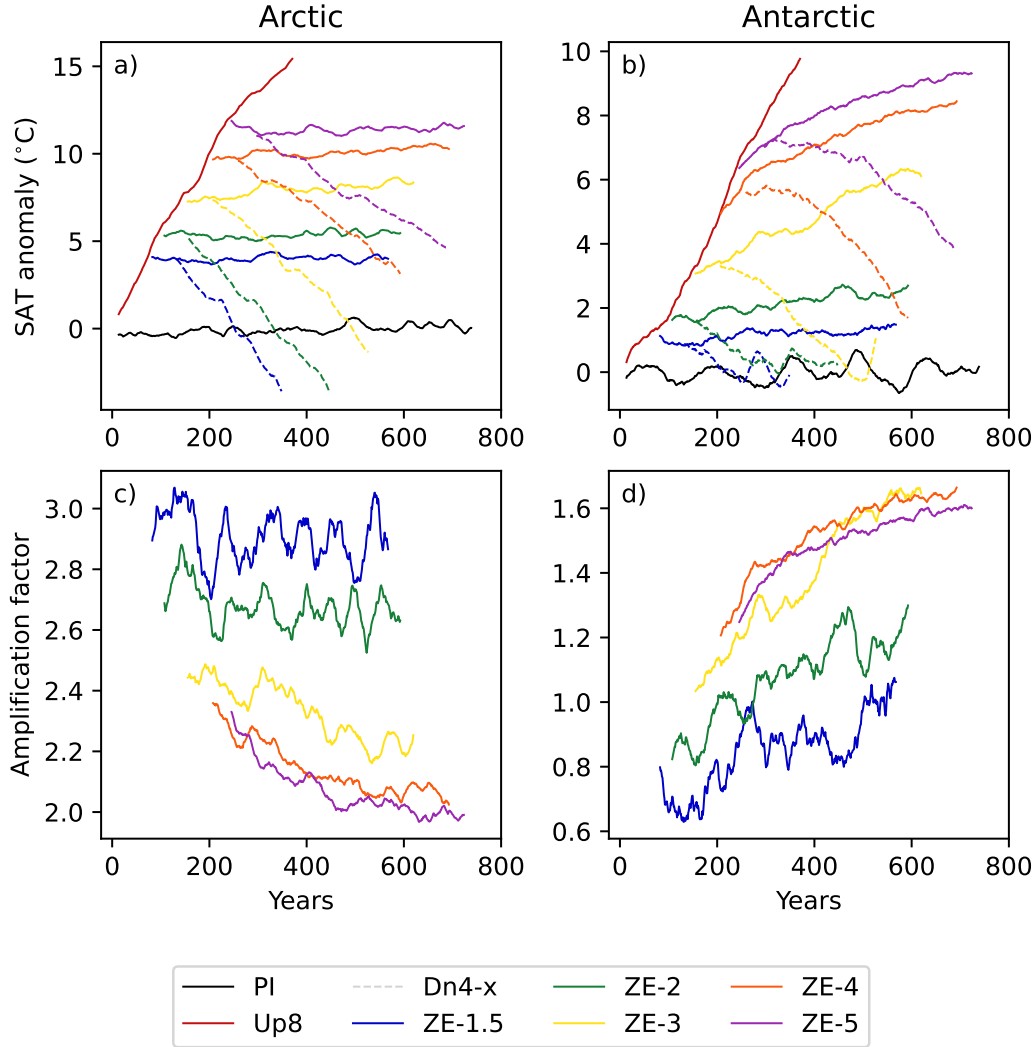

**Figure 2.** Time evolution of high latitude zonal average surface air temperature anomalies (SAT) with respect to the mean of PI. a) and c) show 60-90°N, b) and d) show 60-90°S. c) and d) show the evolution of high latitude SAT anomalies divided by the global mean anomaly for the ZE experiments. All quantities are smoothed with a 30-year running mean. Solid lines show PI, Up8 and ZE-*x* simulations, dashed lines of the same colour show Dn4-*x* cooling simulations from each GWL.

in the Ross and Amundsen Seas, likely caused by a release of ocean heat that has built up below the surface during the warm phases of this simulation.

The polar amplification of GSAT anomalies is not zonally uniform with latitude, and the factor by which warming is increased is significantly reduced over the GrIS and AIS themselves. This is because even at high GWLs the maximum surface air warming on the ice sheets is always constrained by the melting temperature of the ice surface. These factors especially limit





temperature change over Antarctica, resulting in initial polar amplification factors less than one for high southern latitudes
(Fig. 2d). The requirement for latent heat in melting means that surface temperature alone is not sufficient to characterise the change in the energy budget in ice sheet areas.

## 3.2   Arctic sea-ice

During Up8, Arctic sea-ice area declines approximately linearly with global temperature change (Fig. 3), as reported in previous studies (e.g. Ridley et al., 2012). This result is expected because GSAT is closely related to cumulative emissions during our
simulations, and there is an observed linear relationship between cumulative anthropogenic $CO_2$ emissions and Arctic sea-ice decline (Notz and Stroeve, 2016).

The ZE-*x* simulations generally show a rapid stabilisation of Arctic sea-ice once emissions are halted (Fig. 3a,b). This is in contrast to previous studies in which sea-ice area continues to decline during stabilisation runs following the emission phase (Ridley et al., 2012; Li et al., 2013). In ZE-5, the highest GWL in our ensemble, the sea-ice appears to recover by approximately
2.5 million $km^2$ before sequestration starts, suggesting that there are sea-ice restorative feedbacks in response to large-scale melt or global temperature increase. This reversibility of sea-ice decline is consistent with Tietsche et al. (2011) which finds that Arctic sea-ice recovers from dramatic perturbations imposed in the form of a summer sea-ice free state.

The Dn4-*x* simulations highlight the reversibility of Arctic sea-ice with global mean temperature (Fig. 3b) and cumulative emissions. Ramp-down simulated sea-ice area tends to follow a reversed trajectory to that of the ramp-up simulations for all
GWLs.

## 3.3   Greenland Ice Sheet

There is a clear forced signal of Greenland mass loss in the zero-emission runs at all GWLs in the ensemble, compared to the background drift of this ice sheet under constant preindustrial forcing in ZE-0 (Fig. 4). As $CO_2$ concentrations increase, so does the rate of mass loss, with ZE-*x* mass change curves appearing to split from the Up8 curve as local tangents at the time
that emissions are zeroed (Fig. 4a). Stabilising GSAT by zeroing $CO_2$ emissions does not stop mass loss from the GrIS; rather, it simply holds the rate of mass loss fixed at that level. At each GWL there is an approximately linear rate of ice loss that is set by the value of the rate when emissions cease. At a GWL of 1.5°C this amounts to a 0.2 mm/yr sea level equivalent volume (SLE) loss, whilst at 5°C the GrIS loses 2 mm/yr SLE. Like some other variables (e.g. ocean heat content, see section 3.7) there is a considerable time lag between the start of $CO_2$ removal and a change in the sign of the mass trend for the GrIS. In
Dn4-1.5, the forced GrIS contribution to global mean sea level (GMSL) only stops after approximately 100 years of negative emissions, whilst for Dn4-3 it takes approximately 200 years of negative emissions, with even longer stabilisation timescales for the GMSL contribution at higher GWLs. In our ensemble, however, there is no clear evidence of abrupt acceleration of mass loss from the GrIS as a whole, or that any state is reached from which cooling of the climate does not initiate the reversal of GrIS change. It should be noted that the centennial timescales of our simulations are shorter than those of previous studies
in which irreversible loss of the GrIS was assessed (e.g. Robinson et al., 2012; Gregory et al., 2020).





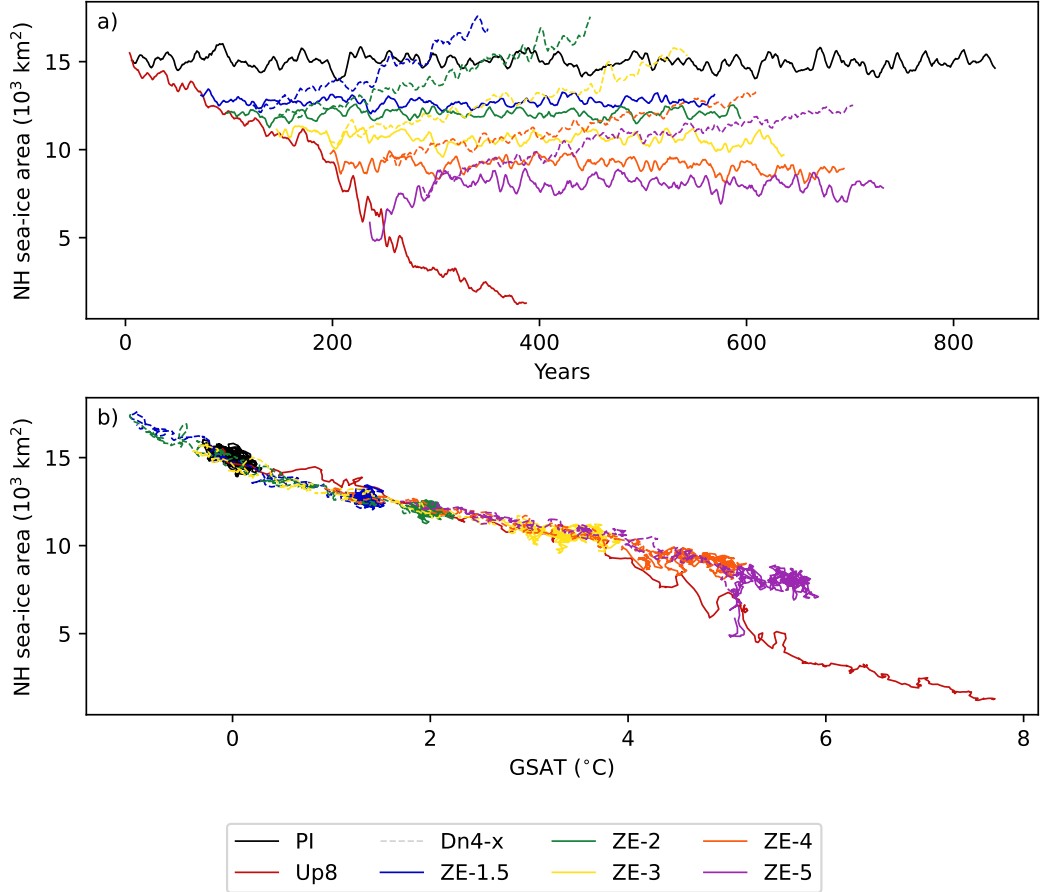

**Figure 3.** Northern hemisphere annual maximum sea-ice area plotted against a) time and b) global average surface temperature anomaly. All quantities are smoothed with a 5-year running mean. Solid lines show PI, Up8 and ZE-*x* simulations, dashed lines of the same colour show Dn4-*x* cooling simulations from each GWL.

The evolution of the mass of the GrIS over the next hundred years or so is expected to be largely controlled by the surface mass balance (SMB, i.e. the difference between surface accumulation and ablation) it experiences rather than dynamical change (Goelzer et al., 2020). There is a clear correlation across our ensemble between the rate of GrIS mass loss and the SMB, supporting this expectation. SMB becomes increasingly negative as GSAT increases and settles to a constant rate when GSAT

stabilises, with an apparently similar form to the relationship derived by Fettweis et al. (2013). Increases in air temperature are associated with greater precipitation which can offset some of the loss of mass from increased melt, but melting increases more quickly with temperature than precipitation and at higher GWLs a significant proportion of the precipitation over the ice sheet falls as rain which instantly runs off in our model. SMB responds rapidly to reductions in GSAT in the Dn4-*x* simulations with the trajectory of GrIS SMB simply reversing, even when reducing from the highest GWLs (Fig. 4b). However, GSAT needs to




be brought back close to PI values before SMB becomes net positive again, and the ice sheet has the possibility of regaining mass, depending on its rate of dynamic discharge.

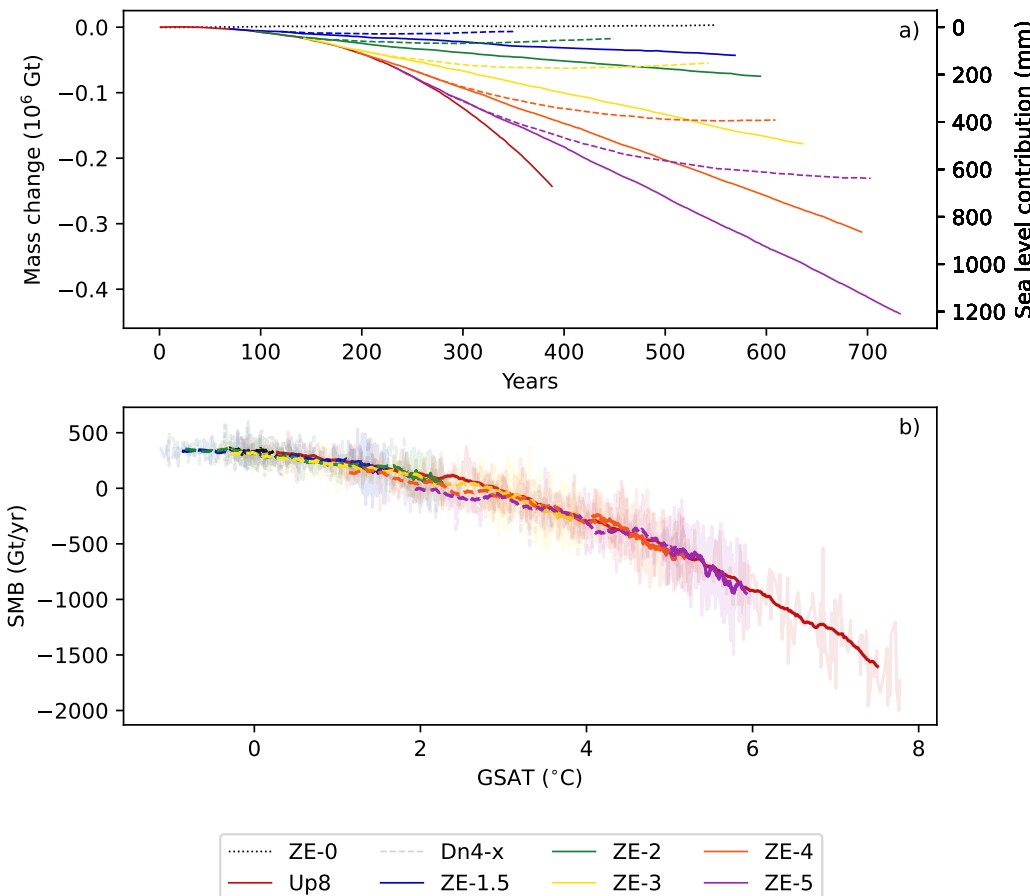

**Figure 4.** Evolution of a) Greenland ice sheet mass with time and b) surface mass balance (SMB) with the global mean surface air temperature anomaly. The annual SMB values are shown with a smoothed trend in bold overlaid. Solid lines show Up8 and ZE-*x* simulations, dashed lines of the same colour show Dn4-*x* cooling simulations from each GWL. The dotted line shows the evolution of the ice sheet under a continuous PI forcing (ZE-0).

The GrIS dynamically responds to this change in surface forcing in UKESM with a characteristic pattern. Ablation at the low-lying, warm edges of the ice sheet thins and slows the ice, which both act to reduce the discharge of solid ice at much of the boundary, especially in the west and south. In UKESM1.2, GrIS marine-terminating glaciers do not directly feel the

evolving ocean boundary conditions because the necessary fjord regions cannot be resolved by the ocean model, so we do not expect to see the acceleration of these glaciers or the associated increase in discharge in those regions in our simulations. This is a limitation of the model because, in reality, ocean-forced changes in ice discharge are responsible for a significant amount





of the current observed ice loss (Mouginot et al., 2019), although in the future ocean-forced mass loss is expected to be small in comparison to SMB (Goelzer et al., 2020).

## 3.4 Antarctic sea-ice

The Up8 simulation shows an approximately linear decline of Antarctic sea-ice with increasing GSAT (Fig. 5), similar to that of the Arctic. In our zero-emission runs, however, with the exception of the ZE-1.5, Antarctic sea-ice continues to decline despite the stabilisation of global mean temperatures. This is in contrast to the Arctic, a regional disparity that is also reported by Lacroix et al. (2024). In the Antarctic, we find that the rate of sea-ice decline per degree of global mean surface temperature warming is increased during zero-emission runs when compared to the Up8 simulation, as reported by Ridley et al. (2012). This indicates that lags in the model Earth system, for example in local surface air temperature, continue to cause sea-ice melt in the stabilisation simulations beyond that caused by the increase of GSAT.

The Dn4-*x* simulations highlight the reversibility of Arctic sea-ice change with respect to GSAT and cumulative emissions (Fig. 3), but the Antarctic response differs, displaying a significant lag in the relationship (Fig. 5). Antarctic sea-ice continues to decrease in the ZE-*x* runs, and when plotted against GSAT the sea-ice in Dn4-*x* simulations does not recover along the same curve as Up8 since it start to recover from a lower ice extent (especially above the 3°C GWL), likely due to the accumulation of sequestered heat in the Southern Ocean. Li et al. (2013) suggested that Antarctic sea-ice change is reversible on longer timescales, with sea-ice lagging surface temperature change by several centuries. This appears to be consistent with our simulations, in which sea-ice only begins to recover in the Dn4-4 and Dn4-5 simulations after 150-200 years.

## 3.5 Antarctic westerlies

One of the major components of the climate system at high southern latitudes is the belt of mid-latitude westerlies that encircles the Antarctic continent, referred to as the 'westerly jet'. It interacts strongly with not only the Southern Ocean, but also parts of the Antarctic continent as its poleward flank extends over important coastal zones (Goyal et al., 2021). The broad behaviour of the southern hemisphere westerly jet can be captured using two indices based on lower-tropospheric zonal mean westerly wind: jet latitude index (JLI) and jet speed index (JSI) as described in Bracegirdle et al. (2017).

There is a general poleward shift of the westerly jet during Up8, which is maintained during the 50-year zero emission periods, followed by a return towards lower latitudes during $CO_2$ removal phases (Fig. 6a). The simulations exhibit broad reversibility, which is apparent across the range of GWLs shown.

The jet speed strengthens as global temperatures rise during Up8 (Fig. 6b), but this strengthening reduces as GSAT increases beyond 4°C. During the periods of zero emissions these enhanced speeds are not in general maintained. Instead, the jet gradually weakens, particularly at the higher GWLs. During global cooling the jet weakens further and at high GWLs does not recover along the same curve, particularly during Dn4-5. In this case the jet returns to a weaker state than it was initially, which is consistent with the weakened storm track found after $CO_2$ removal by Hwang et al. (2024).

This behaviour of jet diagnostics likely results from competing processes, including low-latitude upper-tropospheric warming, which tends to strengthen the jet and shift it poleward, and high-latitude lower-tropospheric warming, which tends to



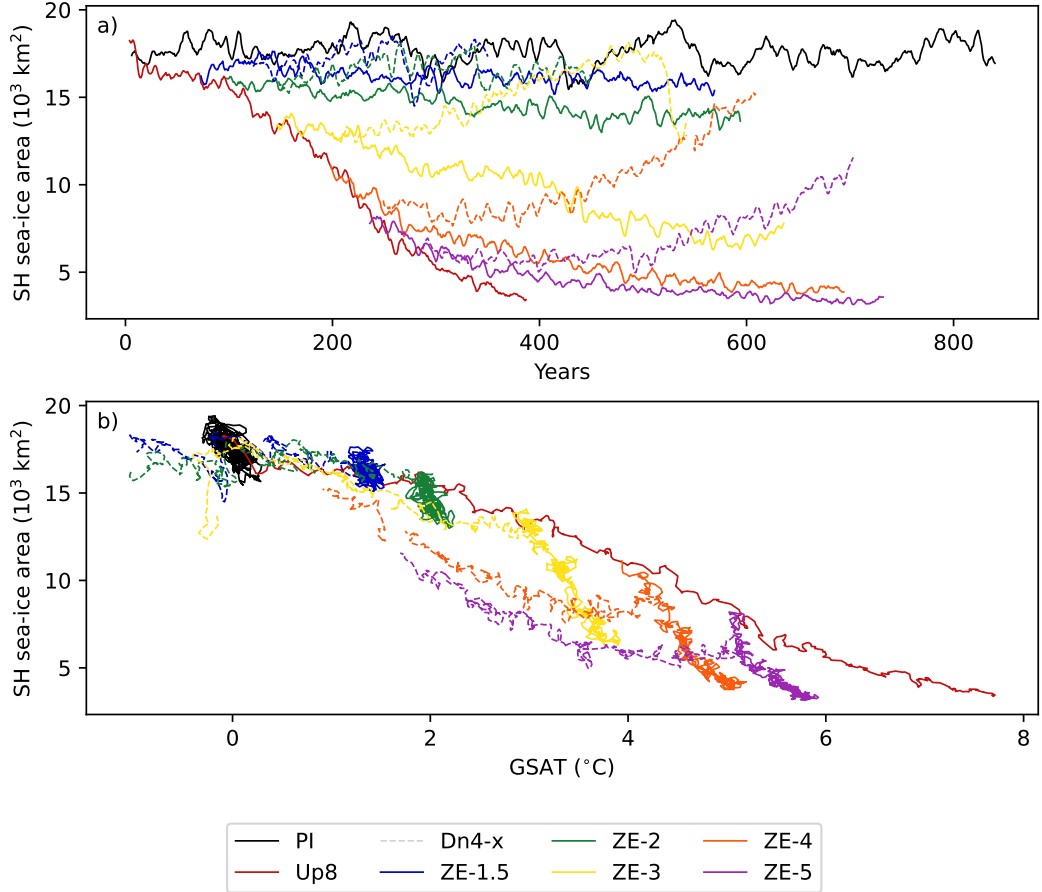

**Figure 5.** Southern hemisphere annual maximum sea-ice area plotted against a) time and b) global average surface temperature anomaly. All quantities are smoothed with a 5-year running mean. Solid lines show PI, Up8 and ZE-*x* simulations, dashed lines of the same colour show Dn4-*x* cooling simulations from each GWL.

weaken the jet and induce an equatorward shift (Bracegirdle et al., 2017; Harvey et al., 2013), as well as interactions with changes in sea-ice area which is shown to also exhibit variations over the ZE-*x* runs (Fig. 5). As suggested by Hwang et al. (2024), a weakening jet during stabilisation at a GWL is consistent with continued SH sea-ice decreases in the ZE-*x* runs. The more prominent lack of simple reversibility in JSI is consistent with the closer link between JSI and sea-ice extent found by

Bracegirdle et al. (2017).

## 3.6  Antarctic Ice Sheet

In all the simulations presented here, the AIS loses mass, providing an increasing flux of freshwater into the Southern Ocean (Fig. 7a). The magnitude of that flux is dependent on the climate forcing, particularly at higher GWLs. However, the majority




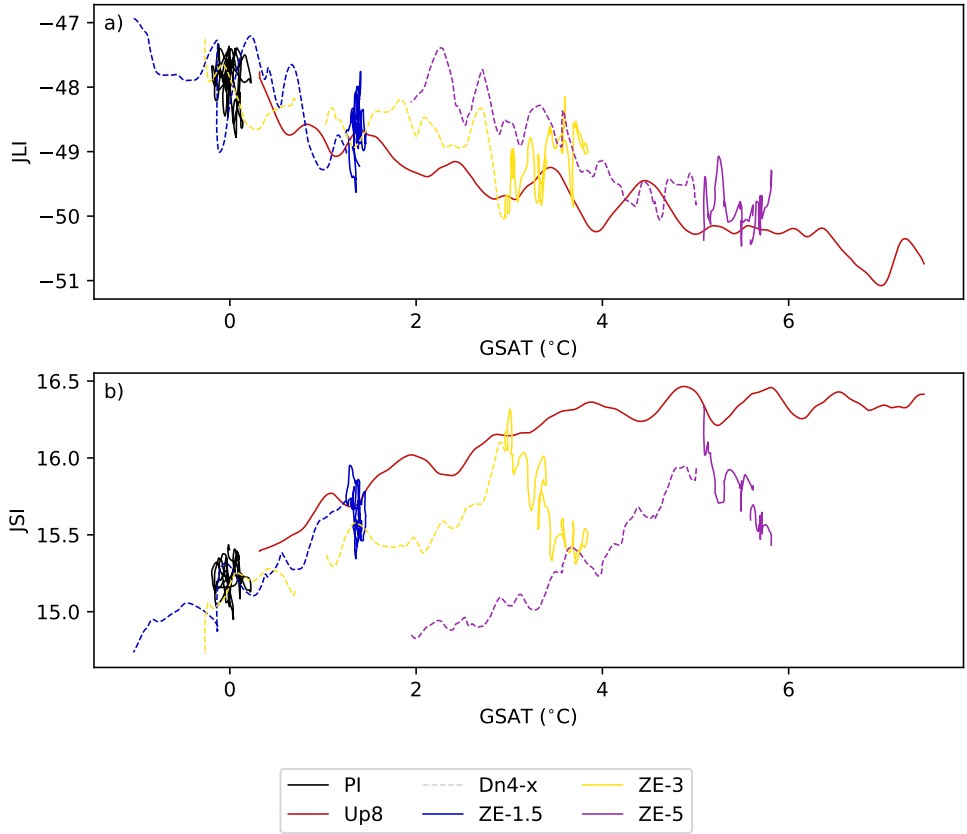

**Figure 6.** Southern hemisphere annual mean westerly a) jet latitude index (JLI) and b) speed index (JSI) for GWLs of 1.5K, 3K and 5K. A 30-year LOWESS smoothing was applied to the jet timeseries before plotting. Solid lines show PI, Up8 and ZE-*x* simulations, dashed lines of the same colour show Dn4-*x* cooling simulations from each GWL.

of this mass loss comes from the basal melting of floating ice shelves, and this part of the mass loss does not directly contribute

to changes in GMSL. To assess the impact on GMSL, we need to examine the change in mass above flotation (Fig. 7b).

Across the range of GWLs, and on the centennial timescales explored in these simulations, the overall change in the ice mass above flotation of the AIS is a result of the interplay between the ice dynamical response arising from the initial condition (as discussed in Section 2.3); changes in the SMB, including snowfall and surface melt; and changes in ice discharge and grounding line position in response to changes in the basal mass balance (BMB) of the floating ice shelves, melted from

290 beneath by the ocean. Over the first 550 years of these simulations, the ZE-0 baseline simulation contributes ~280 mm to GMSL, despite the absence of external climate forcing. This is the result of the unsteady initial condition that leads to mass loss, predominantly from Thwaites Glacier, combined with the ice sheet adjusting to the preindustrial climate state in UKESM when it is coupled. All other simulations presented here contain the imprint of this background ice dynamic behaviour in this sector of the ice sheet. The ZE-0 simulation has the greatest loss of AIS mass above flotation in our ensemble, and therefore





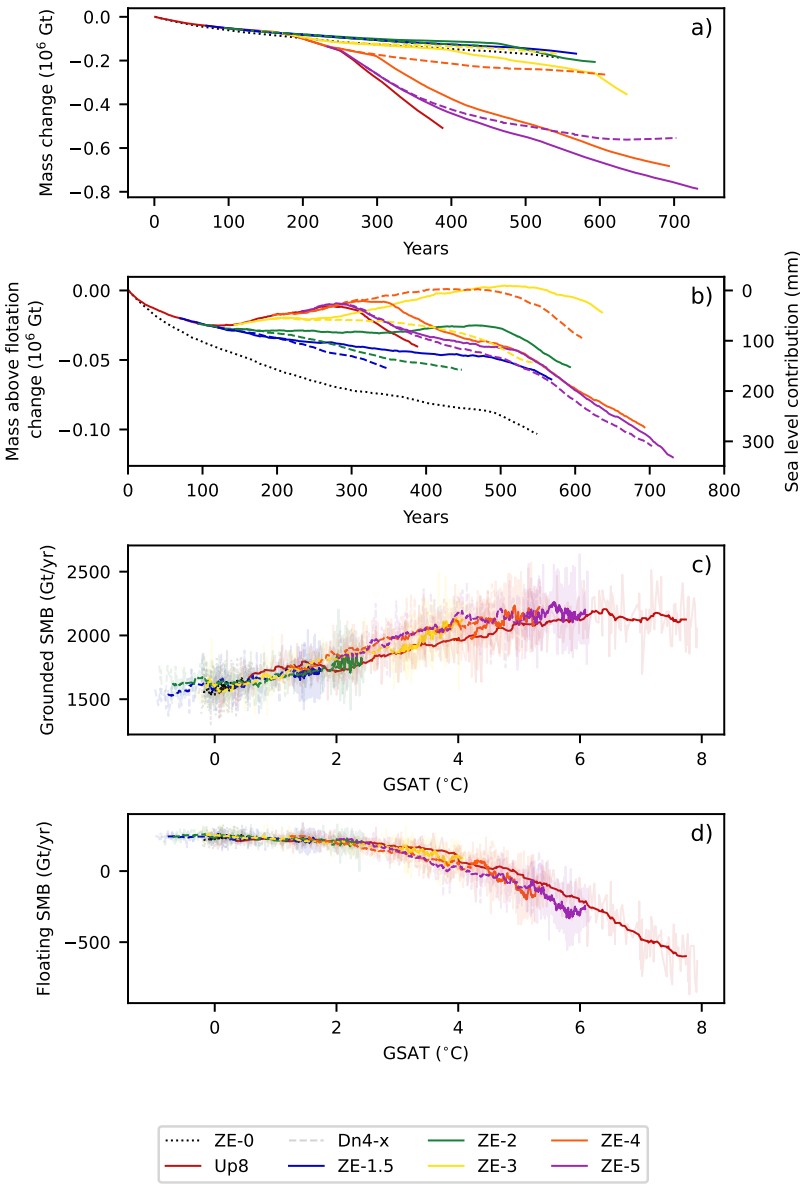

**Figure 7.** Cumulative changes in (a) the mass of the Antarctic Ice Sheet and (b) the mass of ice above flotation in the ice sheet, (c) surface mass balance (SMB) over the grounded ice sheet and (d) the floating ice shelves as a function of global mean surface air temperature. The annual SMB values are shown with a smoothed trend in bold overlaid. Solid lines show Up8 and ZE-*x* simulations, dashed lines of the same colour show Dn4-*x* cooling simulations from each GWL. The dotted line shows the evolution of the ice sheet under a continuous PI forcing (ZE-0).





the largest contribution to GMSL (~280 mm after 550 years). This counter-intuitive result occurs for two reasons: (1) the ice initialisation response, and (2) increased snowfall in a warmer climate. That is, all scenarios lose mass in response to the initial conditions, predominantly from Thwaites Glacier, but the greenhouse gas forcing partially offsets this mass loss by causing increased accumulation of snowfall (positive SMB anomaly) over the grounded regions of the ice sheet. In one simulation (ZE-3), the net GMSL contribution from Antarctica is briefly negative (-2 mm after 550 years), as the cumulative increase in SMB temporarily offsets the dynamic ice loss.

The SMB of the AIS is likely to change in a future, warmer climate, with the primary response on centennial timescales being increased levels of precipitation (in the form of snowfall) from the accelerated hydrological cycle (e.g. Huybrechts et al., 2011). However, quantitative model estimates of the present-day SMB, and projections of future SMB, vary significantly (Mottram et al., 2021). UKESM has previously been shown to project a large increase in Antarctic snowfall compared to other models, which becomes a dominant factor in its simulation of forced change in Antarctic grounded mass balance over the $21^{st}$ century (Siahaan et al., 2022). As with Greenland, our simulations show that SMB changes closely track changes in GSAT and are reversible on similar timescales: SMB stabilises as soon as emissions go to zero and starts to fall as soon as emissions become negative. It is worth noting that the SMB response to increasing atmospheric temperature can be split into two types of behaviour (Kittel et al., 2021): a tendency to decrease in low-lying coastal areas (e.g. ice shelves) and to increase on higher ground further inland. Integrated over the grounded ice sheet, SMB simply increases with temperature due to the dominance of the precipitation component, as the seasonal runoff remains small (Fig. 7c). Quantitatively, the grounded SMB in the ZE-5 simulation increases by ~40% compared to the ZE-0 experiment, and the reduction in grounded SMB in the ramp-down experiments (e.g. in Dn4-4) demonstrates the reversibility of this change as surface temperatures are reduced. On the floating ice shelves, however (Fig. 7d), surface melting begins to play an important role as GWLs increase. At first, this increased melt cancels the impact of the increased precipitation, and then in the ZE-3 to ZE-5 simulations, melt starts to reduce the net SMB over the ice shelves. At the highest GWLs explored, ZE-4 and ZE-5, the SMB integrated over the ice shelves becomes negative, and they experience widespread ablation from both the air above and the ocean below. Where significant surface melting occurs, there is the possibility of hydrofracturing and mechanical breakup of the shelf, depending on the stress regime in the ice (Lai et al., 2020). This effect is not included in our ice sheet model.

The other notable features of the mass loss curves in Figure 7a are the abrupt changes in the gradients due to changes in the ice dynamics. After ~475 years in this set of simulations, there is an acceleration in the loss of mass above flotation that is synchronous across simulations, i.e. independent of the climate forcing (Fig. 7b). This corresponds to Thwaites Glacier entering a period of accelerated thinning and grounding line retreat. The fact that this acceleration also occurs in the ZE-0 experiment suggests that this is an inherent feature of our ice sheet initialisation that takes many centuries to unfold.

Inflection points in the mass loss curves in Figure 7a occur in the Up8, ZE-4 and ZE-5 experiments between 250 and 300 years into the simulations. Here, the driver is a regime shift in the ocean conditions underneath the Ross and Filchner-Ronne ice shelves, as simulated previously by the UKESM for future climate scenarios (Siahaan et al., 2022). Warm water intrudes into the previously cold ice shelf cavities, causing a substantial increase in basal melt rates, reduced ice-shelf buttressing, accelerated upstream ice flow, and grounding-line retreat. The Up8, ZE-4 and ZE-5 experiments switch from a period of



gain in net mass above flotation, to becoming net contributors to GMSL again following the transition (Fig. 7b). A detailed analysis of these rapid transitions between ice-shelf cavity regimes and the impact on AIS mass balance will be presented in a forthcoming publication.

### 3.7 Ocean heat content and global sea level rise

Over the past few decades, the global ocean has stored about 89% of the heat added to the Earth system (von Schuckmann
et al., 2023) and is closely related to the thermosteric component of GMSL. The unique capabilities of the ice sheet coupling in UKESM mean that, for the first time, these simulations allow a systematic investigation of thermosteric and ice sheet contributions to GMSL at different GWLs under zero $CO_2$ emission and subsequent cooling.

Figure 8 shows the time-series of ocean heat content (OHC) evolution for the ensemble, relative to the content at the start of the PI simulation. THe PI has a small (+0.12W/m$^2$) positive top of atmosphere (TOA) budget which leads to a small
accumulation of heat in the ocean under this baseline forcing. For the purpose of our analysis, however, it is the OHC changes arising from the time-dependent forcing that are of interest.

The ocean warms most rapidly in the Up8 simulation but, in contrast to the surface air temperatures, warming does not stop in any of the ZE simulations; rather it persists at a substantial rate and thermosteric sea level rise continues. The warming behaviour in the surface layer is qualitatively similar to that of the full ocean.

In the Dn4-*x* simulations, global ocean warming begins to stabilise with a significant time lag after sequestering $CO_2$. This OHC reaction time scale increases with depth and GWL. In the deepest layer, it takes about 40 years or so for the forcing signal to reach that layer at the start of Up8. Even after 100 years into the Dn4-*x* simulations, layers below 2000m continue to warm. In the top layer the reaction time scale is shorter and the ocean cooling begins more quickly after the start of $CO_2$ sequestration, for example after only 10 years in Dn4-1.5.

Although timeseries of ice sheet mass loss in the simulations are presented in the individual sections above, it is also instructive to view all these components together in a framework that emphasises their relationship to global temperatures (Fig. 9). In this presentation it is clear that sea level rise continues from all of these components under zero $CO_2$ emissions. For the thermosteric component the rate of sea level rise starts to drop, but on the multi-centennial timescales of these simulations this component remains positive throughout (Fig. 9a,d). Sequestering $CO_2$ and reducing surface air temperature does eventually
produce a negative rate of thermosteric sea level rise (i.e. a contribution that acts to reduce GMSL) but this does not even begin to happen until GSAT has reduced significantly. The multi-century timescale of our simulations does not see the thermosteric sea level contribution return to original preindustrial values, even from the lowest GWL in our framework.

The contribution from Greenland is even harder to slow (Fig. 9b,e). Net zero $CO_2$ emission stabilises the rate of loss of the GrIS and this rate does not decline with time as the thermosteric contribution does. Removal of $CO_2$ from the atmosphere
and cooling reduces the rate of sea level rise from GrIS, but only very small negative rates are ever achieved, even once the climate has cooled significantly below the PI baseline. Mass loss from GrIS and its contribution to sea level rise are therefore essentially irreversible in these simulations.





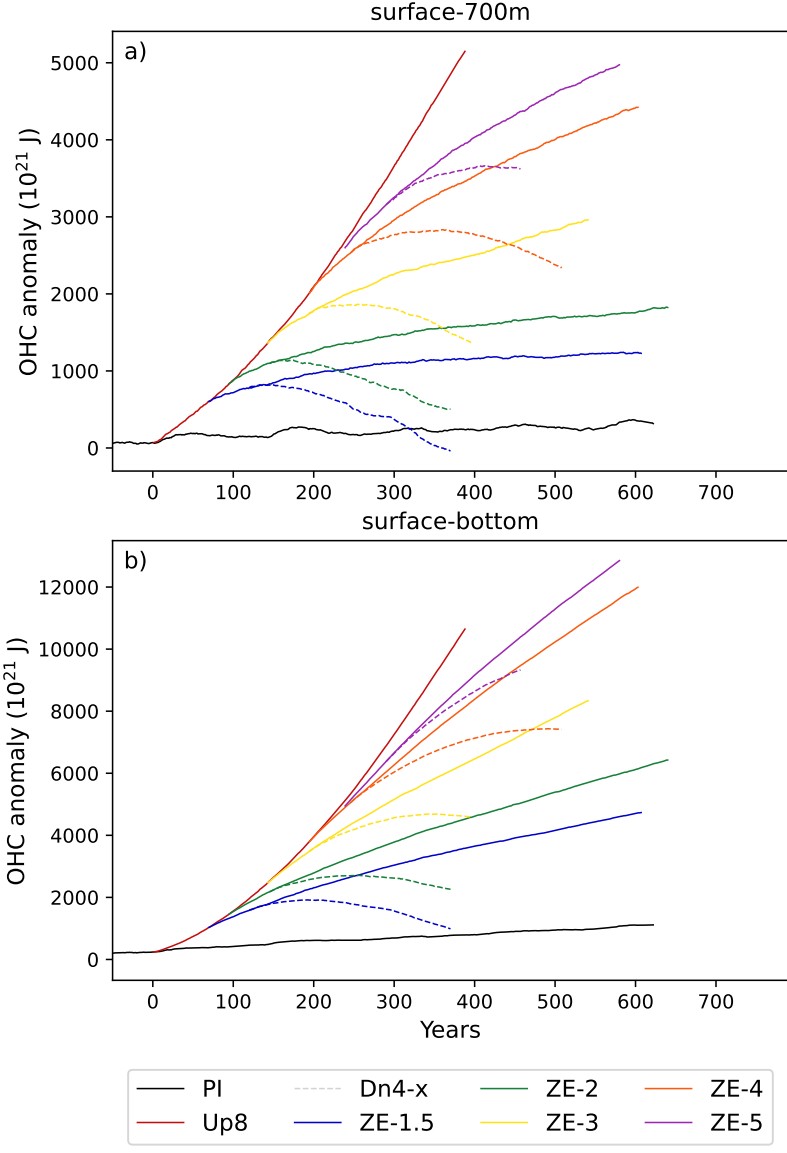

**Figure 8.** Volume average ocean heat content (OHC, ZJ) for a) the top 700m b) full depth of the global ocean. Solid lines show PI, Up8 and ZE-*x* simulations, dashed lines of the same colour show Dn4-*x* cooling simulations from each GWL.

This presentation also highlights the fact that the GMSL contribution from Antarctica should not be viewed as a simple function of global warming level (Fig. 9c,f). More so than Greenland on the timescales of our simulations, the AIS contribution

arises from a complex interaction of the evolving boundary SMB and BMB with the internal dynamics of the ice sheet, some of





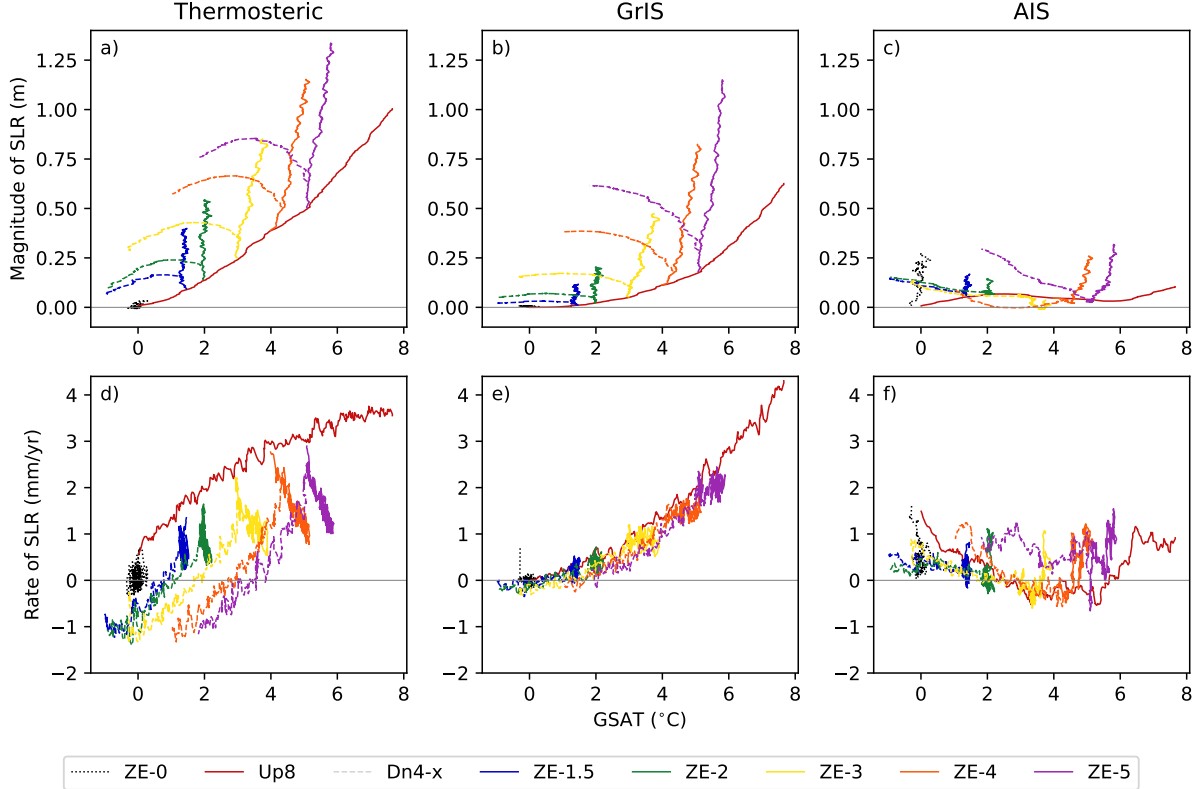

**Figure 9.** Thermosteric and ice sheet contributions to global mean sea level rise (SLR) as a function of global mean surface air temperature (GSAT). Top row shows the absolute magnitude of the contribution, bottom row the amount of SLR per year of the thermosteric (a,d), GrIS (b,e) and AIS (c,f) components respectively. Solid lines show Up8 and ZE-*x* simulations, dashed lines of the same colour show Dn4-*x* cooling simulations from each GWL. The dotted line shows the evolution of the ice sheet under a continuous PI forcing (ZE-0). In this presentation the Up8 simulation line evolves in time to the right, the Dn4-*x* simulations to the left and ZE-*x* simulations are close to vertical.

which are inherent to our ice sheet configuration and initialisation and almost unaffected by the wider Earth system. Indeed, the modern trends in mass loss from the Amundsen sector of the AIS that come with our ice sheet initialisation may be masking a lot of the dynamical response to this climate forcing that might be expected to arise in reality. These are crucial aspects to any decadal or centennial projection of ice sheet behaviour and remain hugely uncertain. Developing techniques for the
initialisation of ice sheet models in coupled climate models is an active area of research.

It is, however, worth noting that one of most impactful aspects of sea level rise is the rate at which it happens, which sets the timescale on which societies must adapt to it. Overall, warming stabilisation and cooling do act to reduce the rate of sea level rise in our simulations, so even if the world is committed to an irreversible magnitude of sea level rise we can still act to control the rate at which this is realised, with potentially important benefits for adaption.





## 4 Conclusion and Outlook

This paper provides an overview of the behaviour of several aspects of the polar climate system in a framework of simulations that allow a systematic exploration of the implications of overshooting, or attempting to stabilise, global warming at different levels. We show that the high latitudes are central to some of the most impactful features of these global simulations: for example, the tendency for GSAT to continue to rise under zero $CO_2$ emissions results from the response of the Southern Ocean. Our analysis also highlights a number of changes in Earth system phenomena that are likely reversible in a warming overshoot scenario where $CO_2$ is sequestered from the atmosphere to reduce surface temperatures, and some that do not simply reverse when temperatures cool. These varied responses across the Earth system and their implications for populations and ecosystems should be considered in any climate policy allowing for greenhouse gas concentrations and surface temperatures to initially overshoot the Paris Agreement targets. An initial study as broad as this can only scratch the surface of the myriad of phenomena and responses. Further, targeted analysis is planned for subsequent papers, where we will consider key process responses and feedbacks in more detail.

These results are from only one ESM, subject to an idealised set of boundary conditions. With those caveats, an assessment of our results suggests qualitative benefits for the polar Earth system from limiting global warming to 2°C above preindustrial levels. Above the 2°C GWL threshold, our simulations show a tendency for GSAT to continue to rise under zero $CO_2$ emissions (Fig. 2), along with a higher degree of sea-ice loss and a lack of recovery of shifts in the winds in the Southern Ocean (Figs. 5 and 6). Cooling back to the PI climate in the ZE-3 experiment leaves a legacy of twice the thermosteric and icesheet contributions to sea level rise than in ZE-1.5 (Fig. 9). These simulations clearly show that there are several aspects of the polar Earth system that are essentially not reversible if temperatures are cooled but, instead, integrate the warming signal they have experienced over time and are slow to release it. In the results shown here an approximate divide between processes in the two hemispheres can be drawn. In the northern hemisphere the Arctic has stronger temperature amplification and correspondingly larger direct responses which are, however, more simply reversible than those in the southern hemisphere. In the south, there is a greater potential to trigger a more complex set of physical interactions with longer timescales which are thus also harder to return to their preindustrial states. For all these aspects of polar change, we show that it is worth following greenhouse gas mitigation policies in which every amount of additional warming and time it is experienced are kept to a minimum.

The results of ESM simulations such as these are well-recognised to be model dependent. Through the international TIPMIP activity (Winkelmann et al., 2025) a reduced set of the experiments presented here form the basis of a TIPMIP Tier 1 ESM experiment protocol (Jones and al., 2025). This protocol is being actively run by 12 ESMs. Intercomparison of these results will help to determine how robust some of the UKESM results are. A particularly valuable feature of the TIPMIP project is that the ESM simulations will be used as boundary conditions to force ensembles of a wide range of models of different Earth system components, capable of investigating the impact of climate warming, stabilisation and recovery in their own specialised domains in much greater depth than is possible in global ESMs.

Other CMIP7 MIPs such as ScenarioMIP (van Vuuren et al., 2025), flat10MIP (Sanderson et al., 2024) and CDRMIP (Asaadi et al., 2024) are conducting investigations into the impacts of climate stabilisation and overshoot using ESM forcing scenarios

that are more complex than the $CO_2$-only framework used here. Other anthropogenic climate forcers such as aerosols and ozone play crucial roles in determining very significant regional patterns of Earth system change and need to be taken into account in some regional and near-term projections of real-world climate change. Including such forcers makes it much harder to ensure a common rate of global warming and arrival time at a given GWL (before zero emissions are prescribed) across participating ESMs. This is a key aim of the TIPMIP ESM protocol, allowing a multi-model analysis based on common simulated pathways of global warming, zero-emission (warming stabilisation) and negative emissions (global cooling).

Advanced ESMs such as UKESM include representations of a wide range of Earth system components and as such are valuable tools for gaining an understanding of the possible behaviours, interactions and feedbacks that we might expect to see in the real Earth system when it is subject to time-varying forcing. Features like the interactive ice sheets in UKESM are beginning to enable such models to simulate ever more aspects of the Earth system as native, fully consistent parts of their output. ESMs are starting to include components whose adjustment and equilibration timescales are beyond those traditionally considered in CMIP6 AOGCM simulations. This brings new challenges in determining how such components should be initialised and evaluated. The complex behaviour of the Antarctic component in these simulations is a good reminder that, despite the process-complexity in our current ESMs, there is still much work to be done to improve how they represent key ES processes, including appropriate initialisation, to robustly address key outstanding questions.

*Data availability.* Selected CMORised output from the UKESM TIPMIP ensemble can be publicly accessed at https://gws-access.jasmin.ac.uk/public/ukesm/TerraFIRMA. The full simulation output are archived at the UK Met Office and are available for research purposes through the JASMIN platform (www. jasmin.ac.uk) maintained by the Centre for Environmental Data Analysis (CEDA); for details please contact UM_collaboration@metoffice.gov.uk, referencing this paper.

*Author contributions.* Conception: RSS, CJ, JM Coordinated and conducted simulations: CJ, RSS, SL, JW, SR, MS, JP Analysed output: RSS, KAN, CL, TM, TaB, TK, SL, TJB, AP, RaS, ST Provided manuscript content: RSS, CL, TM, TaB, TK, AO, SL, TJB, RaS, ST Commented on manuscript content: RSS, KAN, CL, PH, CJ, TM, TaB, ThB, AO, TK, JW, TJB Contributed to obtaining funding: RSS, CJ, PH, TK, JM, AP

*Competing interests.* The authors declare no competing interests

*Acknowledgements.* Supported by the UK Natural Environment Research Council through the TerraFIRMA: Future Impacts, Risks and Mitigation Actions in a changing Earth System project, Grant reference NE/W004895/1 and The UK Earth System Modelling Project, Grant reference NE/N017951/1.



We acknowledge the project TipESM "Exploring Tipping Points and Their Impacts Using Earth System Models". TipESM is funded by the European Union. Grant Agreement number: 101137673. DOI: 10.3030/101137673. Contribution nr. 30. This work was funded by UK Research and Innovation (UKRI) under the UK government's Horizon Europe funding Guarantee. Grant Agreement numbers: 10090271,10110109, 10121766, 10103098.

We acknowledge the project OptimESM "Optimal High Resolution Earth System Models for Exploring Future Climate Changes". OptimESM is funded by the European Union. Grant Agreement number: 101081193. This work was funded by UK Research and Innovation (UKRI) under the UK government's Horizon Europe funding Guarantee. Grant Agreement numbers: 10043072, 10041076, 10040511, 10038604.




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

## Appendix A: Experiment nomenclature

The simulations presented in this paper are r1i1p1f1 members of the MOHC UKESM1.2 TIPMIP ESM ensemble. For readability we have used a simple approach to experiment naming in this paper, which maps onto the full tipemip_esm naming protocol as follows:


| PI | esm-piControl |
|---|---|
| Up8 | esm-up2p0 |
| ZE-1.5 | esm-up2p0-gwl1p5 |
| ZE-2 | esm-up2p0-gwl2p0 |
| ZE-3 | esm-up2p0-gwl3p0 |
| ZE-4 | esm-up2p0-gwl4p0 |
| ZE-5 | esm-up2p0-gwl5p0 |
| Dn4-1.5 | esm-up2p0-gwl1p5-50y-dn1p0 |
| Dn4-2 | esm-up2p0-gwl2p0-50y-dn1p0 |
| Dn4-3 | esm-up2p0-gwl3p0-50y-dn1p0 |
| Dn4-4 | esm-up2p0-gwl4p0-50y-dn1p0 |
| Dn4-5 | esm-up2p0-gwl5p0-50y-dn1p0 |