# Peer review of "Response of ice sheets, sea-ice and sea level in climate stabilisation and reversibility simulations using a state-of-the-art Earth System Model"

_EGUsphere, 2025_

## Author Comment (AC1)

We thank both reviewers for their time and the helpful comments which we address in turn, below. In general, we are confident that we can provide a revised manuscript that will satisfy their concerns and be appropriate for final publication in Earth System Dynamics.

**Reply to Reviewer 2:**

Smith and co-authors present the first description of new idealized Earth system model simulations conducted with the UKESM, which is used here in its fully coupled configuration, including dynamic Greenland and Antarctic ice sheets. The simulations encompass the Tier 1 experiments of the Tipping Point Model Intercomparison Project (TIPMIP) but extend substantially beyond them by incorporating additional global warming levels and branched simulation pathways. Running a model of this complexity, particularly with fully dynamic ice sheets, is a significant achievement requiring expertise across multiple disciplines. Nevertheless, the model setup remains somewhat unsatisfactory, as the ice sheets are initialized directly from present-day conditions, resulting in noticeable model drift even in unforced scenarios. The authors acknowledge these limitations and point to ongoing research aimed at addressing them.

The manuscript is well written and provides a concise overview of the simulations, with a particular emphasis on high-latitude processes. However, it remains largely descriptive and does not dive deeply into underlying mechanisms or dynamics. This is also reflected in the figures, which primarily present simple time series and omit visualizations of the more complex dynamics governing these overshoot scenarios. Given this, I wonder whether the manuscript may be more appropriately suited to *Earth System Science Data* rather than *Earth System Dynamics*.

Thanks for reviewing and the positive comments. As we state, our group has a suite of papers under review and in progress with more detailed analysis of individual aspects of these simulations but we think an overview paper such as this is an important introduction and overview in its own right. Although we acknowledge that it does not go into great physical detail of the results, our paper does include analysis and interpretation of the simulations and we feel that it is better suited to ESD than ESSD.

Beyond these general comments, I have two major points that I would like the authors to address:

1. Definition of reversibility.

The manuscript refers repeatedly to reversibility and irreversibility of sea ice, ice sheets, and other components, based on the path-(in)dependence of variables during ramp-up versus ramp-down phases. However, these responses are highly transient and differ fundamentally from the traditional concept of reversibility associated with hysteresis, which concerns (quasi) steady-state behavior. I encourage the authors to provide a more thorough definition of reversibility, one that explicitly accounts for the transient nature of the experiments and the characteristic internal timescales of different components (e.g., sea-ice versus ice-sheets).

This is a very good point. An earlier version of our paper did in fact include a discussion of exactly this issue which was cut for the sake of brevity. We will gladly take the opportunity to reinstate some of the material that addresses this topic.

2. Focus on high latitudes despite ad-hoc ice-sheet initialization.

The strong emphasis on high-latitude and ice-sheet responses is difficult to reconcile with the highly idealized ice-sheet initialization, which the authors themselves acknowledge as a major

limitation. This is not a criticism of their general approach; coupling dynamic ice-sheets within fully complex ESMs remains extremely challenging due to their large inertia, long equilibration timescales, and the high computational cost of such simulations. However, given the substantial biases introduced, particularly for the Antarctic ice sheet, it is unclear whether the resulting ice-sheet behavior can be meaningfully interpreted. As there is no straightforward solution to the initialization problem, I recommend reducing the emphasis on ice-sheet responses and instead providing a more detailed analysis of other Earth system components, such as the AMOC, ocean circulation in general, or terrestrial changes.

We're glad that the Reviewer agrees with us that there are currently no easy answers to this conundrum. However, we believe that there are nevertheless scientifically robust conclusions that can be drawn from the behaviour of the ice sheets in these simulations with careful analysis, and do not believe we have gone beyond those limits here.

As part of addressing a related point by the other Reviewer we will include both PI and ZE-0 lines on all figures to more explicitly demonstrate that the background trend in ice sheet evolution is not significantly biasing the climate simulation. These figures will also demonstrate in particular that the trend does not affect ice sheet surface mass balance (SMB) which is an extremely important aspect of the ice sheet simulation and comprises half of the paper's material focusing on ice sheets.

For the GrIS, in section 3.3, the forced signal of mass change is clearly very much larger than any background trend, so we think that GrIS mass change can be considered as a robust response to the experimental protocol. Our simulated AIS mass change is clearly something that needs to be interpreted with more nuance. We see no significant trend in either SMB or basal mass balance (BMB) for AIS under the PI climate. The most apparent background trend attributable to our initialisation is that of the Mass Above Flotation (MAF) visible in Fig 7b. As noted in the text, this is attributable to the continuation of the currently-observed thinning of Pine Island and Thwaites Glaciers. Although significant in terms of the MAF of the ice sheet, this background trend is independent of GWL, relatively constant for all except the last few years of the simulations and only affects a geographically very-limited part of the ice sheet. This regional MAF trend does not significantly affect mass loss from the large floating shelves of the Ross and Filchner-Ronne which is the largest part of the forced mass loss of the AIS as a whole in this model, on these timescales (compare FIg 7a with 7b, especially the vertical scales), nor does is interact significantly with the AIS-integrated SMB which is the other key part of the whole-icesheet mass balance.

So, to the degree that we do interpret the forced evolution of the AIS, we believe what we have said and shown in this paper is physically meaningful. We certainly agree that there is a long way to go to develop more satisfactory ways of initialising ice sheet components for coupled ESM simulations, and that this significantly limits how we can relate these simulations to projections for the real world, particularly when considering their change in MAF and contribution to sea level rise. It is for this reason that in section 3.7 we go no further than concluding that

"the GMSL contribution from Antarctica should not be viewed as a simple function of global warming level. More so than Greenland on the timescales of our simulations, the AIS contribution arises from a complex interaction of the evolving boundary SMB and BMB with the internal dynamics of the ice sheet"

As for refocusing the paper to include more parts of the Earth System in the description, we must once again defer this to other parts of our wide group who have their own papers in progress analysing exactly these things, which we are not really at liberty to include here.

Minor comments:

L2: Please mention the range of GWLs here.

OK

L11: Mention after how many years of zero CO2 emissions, CO2 was removed again from the atmosphere.

OK

L67: branched off instead of spawned?

OK

L73: I would prefer to have Appendix A moved to the main text, to further improve readability and look up of simulation names.

OK

L96: Can you briefly explain why the negative emission rate was set to half the positive one and not the same?

Our wider experiment protocol (which Reviewer 1 would rather we say less about) includes testing the sensitivity of Earth System response to carbon sequestration at different rates. Using the same rate of negative emissions as positive ones led to some stability problems in UKESM, so in purely practical terms we got most simulation output for analysis, most quickly, from the half-rate experiments. For the level of detail and conclusions present in this overview paper, the rate of negative emission is not significant. Since Reviewer 1 would like to see less mention of the wider protocol for simulations not used in this paper, we will try to balance these two requests in the revised paper

L115: Is JULES also running on a 1° grid?

The JULES base longitude-latitude grid is identical to that of the atmosphere, (1.875∘ x 1.25∘). We will note this in the revised text

L120: Please briefly mention the solver for the ice-sheet dynamics (SIA, SSA, hybrid …)

We use the L1L2 solver (Cornford et al. 2013). We will note this in the revised text

L142: Are the ice-sheets in thermal equilibrium at time of branching?

We don't have the thermodynamics component activated in these simulations. The ice sheets maintain their initial internal temperature and effective viscosity fields throughout. We will note this in the revised text.

L295: Explicitly mention again that this refers to the first 550 yr of the simulation.

OK

L339: Typo: THe PI

We will fix this

Fig. 1: Can you add a panel (or a separate figure) depicting the AMOC evolution for these different model runs? In panel c, why is there an offset at the beginning of the simulations of PI and Up8?

We are reluctant to do this, as others in our group are preparing a paper solely focussing on the AMOC evolution in these simulations. It is a substantial topic in its own right, and we would rather not briefly allude to it here without doing it justice.

We think the offset between black and red lines in Fig 1c results from the 30 year running mean that has been applied to the timeseries, meaning that nothing can be plotted for the first 15 years of data for PI and Up8. This does highlight however that there appears to have been a problem applying the equivalent smoothing in panel b), since they do not have offsets. This will be checked and corrected in the revised manuscript.

Fig. 7: I find the term Mass Change slightly confusing here. Initially, I thought this refers to the net mass change, instead of the loss term (correct?).

Mass change refers to the cumulative, net mass change of the ice sheet. E.g. in 7a, negative numbers mean less ice than at the start of the simulation. 7a refers to the whole ice sheet, grounded ice and floating shelves combined, and 7b shows only the change in ice mass that would contribute to sea level rise (ie largely excluding the shelves) via the Mass Above Flotation definition. We will clarify this in the revised text.

---

## Author Comment (AC2)

We thank both reviewers for their time and the helpful comments which we address in turn, below. In general, we are confident that we can provide a revised manuscript that will satisfy their concerns and be appropriate for final publication in Earth System Dynamics.

**Reply to Reviewer 1:**

The paper describes an ensemble of simulations investigating the effect of stabilisation of CO2 levels at different global temperature anomalies using a coupled ESM/ice sheet model for Greenland and Antarctica including a scenario mimicking the effect of removal of the entire anthropogenic CO2. The experimental design follows the TIPMIP protocol and fills in additional levels at which stabilization takes place. The topic is highly actual and fits perfectly into ESD. In general, the paper is well written. The analysis could go more into depth, but the authors see this as a first paper, introducing the set of experiments and leaving the detailed analysis for later paper.

Thanks for reviewing and the positive comments. As you allude to, our wider group has a suite of papers in progress and under review with more detailed analysis of individual aspects of these simulations. We think an overview paper such as this is an important introduction and overview in its own right.

It is sufficient material to be published, but in some places not enough analysis to make me really happy. One example: Following your argument in section 3.6, the Antarctic SMB closely follows the GSAT (306/7) and stabilizes as soon as emission stop (307). This is obviously differently then the behavior of Antarctic SAT, which continues to warm even after stopped emission (Fig.2b and discussion). Here a bit more careful discussion and analysis of the different behaviors after stabilization and the causes behind would be essential. Why is the local Antarctic SAT not relevant?

This example is a good point and highlights an explanation we should have been more careful with. The local SAT over the ice sheet is indeed more relevant to SMB than either global SAT or emissions. In the case of higher GWLs, as noted elsewhere, GSAT continues to rise when emissions stop, as do local AIS SAT and SMB with it. At lower GWLs, where GSAT does basically stabilise when emissions cease, the high latitude SAT shown in figure 2 (the average of the whole 60-90S region) has a much smaller warming signal and the local AIS SAT itself has further regional variation, making it harder to detect and interpret a robust trend in continentally-integrated SMB. In all cases SMB is best correlated with the local SAT over the icesheet itself, and the relationship between the local SAT, global SAT and the cessation of global carbon emissions is more complicated than we implied at line 306. In a revised draft we will remove the line in question and clarify this point with additional explanation.

In general the paper can be published after a bunch of minor corrections.

Some general comments to the figs.:

1. The yellow line is almost invisible in my printed version, on the screen it looks fine. Changing this into orange could be a compromise.

OK

2. In the (mostly temp.) anomaly time series (1cd, 2ab, 4a, 7ab) a zero line should be plotted. That makes it considerably easier to assess potential drift.

OK

3. In some places PI is used as reference, in others ZE-0. This is rather inconsistent. I recommend to plot them both. This would also allow the reader to estimate, whether the drift in the ice sheet has an effect on the southern ocean climate (sea ice, temp) or not.

We will do this. Although we don't believe this will show significant differences, we think it is a good suggestion for clarity and to explicitly demonstrate that the drift in ice sheet evolution under the PI forcing does not have a major effect on the climate state, a concern also noted by reviewer 2.

Detailed comments

37 crucial ?that?

OK

62 specified emission of CO2 be more specific and give the number

OK

96-98 specify the length of the runs

OK

100-104 does not make sense to describe experiments specifically that are not used in the paper. Here a vague hint to more experiments should be sufficient

We note that Reviewer 2 would like an explanation of the choice of negative emission rate, which is most sensibly done by reference to the wider set of experiments. We will try to balance these two requests in the revised paper.

187 southern hemisphere SAT is inaccurate, you are discussing only the polar SAT. The PI runs shows similar multidecadal variability

It is true that in this view the event in question does not look so much larger than those seen in the warmest phases of PI variability. However, the event in question is exceptional since it rebounds to this level of warming despite the global SAT and pCO2 being lower than PI at this point in Dn4-3, so it is clearly worthy of comment. The comparison to PI variability is useful however, and we will expand on the similarities and differences between this event and those oscillations in the revised text.

192-196 max surface air warming .. is always constrained by the melting temp of the ice surface..

While I can follow this argument for Greenland and higher CO2 levels, where it is at least true in summer, this is almost completely irrelevant for Antarctica. Even in the highest scenarios the melting is restricted to coastal areas. The high elevation areas of the ice sheet are and will be far away from the melting temp and are obviously accumulating happily mass (see fig.7c). Give a better reason!

This is another point we should have been more careful with, and will expand on in the revised manuscript. There is literature around how the polar amplification signal differs between the northern and southern hemispheres, and especially how relevant heat transport and radiative feedbacks are influenced by the topography of the Antarctic ice sheet to limit warming (eg Salzmann 2017, ESD) that explains this feature more physically.

200 does the physics of sea ice depend really on the cumulated emissions or rather the Arctic SAT, which is linearly related to GSAT and the cumulated emissions? Please give a physically plausible reasoning!

We're not sure about this comment. We agree that the sea-ice state is physically related to local SAT, and that this in turn is correlated with GSAT and cumulative emissions. In this paragraph we state a correlation evident in figure 3 between our sea-ice simulated sea-ice area and GSAT, and that seems reasonable to us. The statement on line 200 that appears to be being queried however is "there is an observed linear relationship between cumulative anthropogenic CO2 emissions and Arctic sea-ice decline"  - this is not our reasoning, here we are simply repeating the clearly stated conclusion of Notz and Stroeve (Science 2016) which supports the correlation we see in our simualtion. In the revised manuscript we will be more clear that it is not the increased emissions themselves that directly cause melt, rather the increase in SAT and heat content of the polar regions that does so.

220 How does the sign change of the GrIS mass contribution relate to the time, when the GSAT anomaly becomes negative?

This is an interesting question (closely related to the later comment about line 362), and one we will take the opportunity to expand on in the revised manuscript. Figure 4 shows that the gradient of the relationship between SMB and GSAT when the GSAT anomaly is near 0 is almost flat, and figure 9 shows that the GrIS mass contribution only barely becomes negative even in the Dn4-1 and Dn4-1.5 simulations where both the Arctic and GSAT temperature anomalies have dropped significantly below the PI level. We will include some analysis of the ice sheet behaviour at below-PI GSATs following carbon sequestration in the revised manuscript.

section 3.6 The effect of the residual mass loss on the climate is not shown at all. If it is negligible, great, than please explicitly state this. Showing ZE-0 also in the climate plots particularly in the south would remove remaining doubts.

We will do this, and as noted above will follow the suggestion of plotting both PI and ZE-0 lines on all figures.

339 THe typo

We will correct

362 here or somewhere else a small discussion would be helpful, that GrSMB does not lead to more ice production for negative GSAT anomalies (Fig. 7d). Please discuss the mechanism(s) behind this.

see our reply above re: line 220. We will include some analysis of the ice sheet behaviour at below-PI GSATs following carbon sequestration in the revised manuscript.